# Ionization Cross Sections of Hydrogen Molecule by Electron and Positron Impact

**DOI:** 10.3390/ijms25063410

**Published:** 2024-03-18

**Authors:** Károly Tőkési, Robert D. DuBois

**Affiliations:** 1HUN-REN Institute for Nuclear Research (ATOMKI), 4026 Debrecen, Hungary; 2HUN-REN Centre for Energy Research, 1121 Budapest, Hungary; 3Department of Physics, Missouri University of Science and Technology, Rolla, MO 65409, USA; dubois@mst.edu

**Keywords:** classical trajectory Monte Carlo model, ionization, charge exchange cross sections, electron impact processes, positron impact processes

## Abstract

We present ionization cross sections of hydrogen molecules by electron and positron impact for impact energies between 20 and 1000 eV. A three-body Classical Trajectory Monte Carlo approximation is applied to mimic the collision system. In this approach, the H_2_ molecule is modeled by a hydrogen-type atom with one active electron bound to a central core of effective charge with an effective binding energy. Although this model is crude for describing a hydrogen molecule, we found that the total cross sections for positron impact agree reasonably well with the experimental data. For the electron impact, our calculated cross sections are in good agreement with the experimental data in impact energies between 80 eV and 400 eV but are smaller at higher impact energies and larger at lower impact energies. Our calculated cross sections are compared with the scaled cross sections obtained experimentally for an atomic hydrogen target. We also present single differential cross sections as a function of the energy and angle of the ejected electron and scattered projectiles for a 250 eV impact. These are shown to agree well with available data. Impact parameter distributions are also compared for several impact energies.

## 1. Introduction

The understanding of the ionization and charge exchange processes in electron/positron–atom and electron/positron–molecule collisions are of fundamental interest in fields ranging from atmospheric and interstellar physics to radiation damage of solids, surfaces, and biological systems. Compared to atomic targets, molecular targets present additional challenges both in experiments and theories due to the many-body character of the collision system. Even for the simplest molecule, H_2_, the development of accurate electronic wave functions is very difficult. In addition to the presence of two electrons, the multicenter nature of the hydrogen molecule introduces difficulties in modeling and calculating various processes. An additional theoretical challenge is performing calculations for lepton impact since there are forces from many target particles that must all be accounted for. A theoretical method that accounts for all these forces is the Classical Trajectory Monte Carlo (CTMC) method. However, modeling and calculating the temporal behavior of many bodies is still a formidable task. Therefore, the study performed here is an investigation of a simplified version of an H_2_ molecule, namely by describing it as a central core potential and a single-bound electron. Although much more sophisticated models exist, an advantage of this CTMC model is that cross sections as functions of many different parameters, with examples being scattering or ejection angles and energies, a correlation between post-collision particles, impact parameter information, etc., can be extracted from a single calculation. These parameters aid in achieving an understanding of collisional processes for the simplest molecule of interest here, which may open the way to investigate more complex and larger molecules.

During the past years, the cross sections for electron and positron impact on H_2_ have been studied extensively, both experimentally [1,2,3,4,5,6,7,8,9,10,11] and theoretically, using various models and methods, such as the R-matrix approach [12], various versions of the distorted-wave Born approximation (DWBA) [12,13,14,15,16,17], and the convergent close-coupling (CCC) method [18]. In addition to these quantum mechanical calculations, classical theoretical calculations have also been used for predicting the cross sections in collisions between electrons and positrons with various targets. Meng et al. developed a classical description for state selective electron capture from H_2_ when all particles are taken into account [19]. Using this classical model, they obtained a fairly good agreement with experimental data. In general, a frequently used classical model is the Classical Trajectory Monte Carlo method. During the last few decades, there has been a great revival of CTMC calculations for atomic collisions involving three or more particles [20]. The CTMC method is useful for treating atomic collision systems involving many body constituents because quantum mechanical methods become very complicated or unfeasible. This is usually the case when higher-order perturbations are required, or many particles are active in the processes. For example, a many-body collision involves a projectile and target nucleus or nuclei, a target electron(s), and possibly a projectile electron(s). An advantage of the CTMC method is that these many-body interactions can be exactly taken into account during the collisions on a classical level by numerically solving the classical equations of motions for each particle [21,22,23,24]. For electron and positron impacts, it has been successfully applied to studying different types of ionization [25,26,27,28,29,30,31,32].

In this work, we present classical simulations based on the Classical Trajectory Monte Carlo (CTMC) method of the ionization and charge exchange cross sections of hydrogen molecules by electron and positron impacts. A three-body approximation is used, i.e., the projectile is one body, and the hydrogen molecule is two bodies, namely a hydrogen-type atom with one active electron bound to the H_2_^+^ ion with an effective binding energy (E_eff_ = 0.567 a.u.) and an effective core charge (Z_eff_ = 1.165). This approximation was shown to be successful in calculating cross-section studies in collisions between Li and H projectiles and H_2_ [33]. A similar approximation was also successfully used by several groups in the investigation of H_2_O, leading to a good agreement between theoretical predictions and experimental data [34,35,36,37].

Calculations for total ionization, specifically single electron removal, were made for impact energies between 20 and 1000 eV and for singly differential cross sections as a function of the post-collision energy and angle for the ejected electron and the scattered projectile at 250 eV. In addition, impact parameter information was obtained.

## 2. Results and Discussions

We performed CTMC simulations for an ensemble of 5 × 10^6^ primary trajectories for each energy to obtain ionization cross sections for the electron and positron impact on a molecular hydrogen target. Calculations were performed for several energies between 20 eV and 1000 eV. A feature of our method is that the impact parameter information is known for each ionization event. Thus, ionization probabilities as functions of the impact parameter can be extracted.

Figure 1 shows these probabilities for electron (blue curves) and positron (red curves)—H_2_ collisions for impact energies of 50 eV, 100 eV, 250 eV, and 1000 eV. The area under the curves is proportional to the total ionization cross section.

The other notable feature is the shift in the peak maxima to smaller impact parameters with an increasing impact energy. The positions of the peak maxima and the shifts are nearly identical for the electron and positron impact. Also, the shape of the curves is similar, although the curves for the electron impact are slightly more symmetric, meaning they can be fitted with Gaussian functions. The peaks for the positron impact are more asymmetric, which is most notable for the 50 eV impact energy, which has a significantly longer tail out to larger impact parameters. Another interesting feature is that while the total ionization cross sections are almost the same at 50 and 100 eV, the probabilities as a function of the impact parameter are completely different. This is true for both electron and positron impacts.

Figure 2 shows the single electron charge exchange probabilities of the hydrogen molecule by the positron impact as a function of the impact parameter for projectile impact energies of 50 eV, 100 eV, and 250 eV. As for ionization, the charge exchange probabilities also show asymmetric peak shapes. The asymmetry decreases with increasing positron energy.

The corresponding ionization cross sections for the electron impact, obtained from Figure 2 by integrating the impact parameter dependent probabilities with respect to the impact parameter, are shown in Figure 3. Included are cross sections we calculated using some additional energies. These are compared with examples of reported theoretical [38,39,40] and experimental [41,42,43,44,45] data. All experimental data are in reasonable agreement with each other. Our three-body CTMC model is in excellent agreement with the experimental and theoretical results from 80 eV to 400 eV. However, for the electron impact, our simple model yields cross sections that are larger at lower energies and smaller at higher energies compared with the experimental data. Our cross-section maximum is approximately 40 eV, whereas the experiment and a more sophisticated theory have a maximum cross section at an energy roughly twice this value. In Figure 3, we also show the scaled experimental ionization cross sections of the atomic hydrogen atom by the electron impact [46], i.e., the atomic cross sections were multiplied by two (see the green solid line).

Figure 4 shows the total ionization cross sections as a function of the impact energy for the positron impact. We can clearly distinguish two main branches of experimental data, which differ from each other below 100 eV. The differences are because the cross sections of Fromme et al. [47] and Moxom et al. [48] contain both ionization and positronium production, i.e., charge exchange processes, while the cross sections of Knudsen et al. [49] and Jacobsen et al. [50] contain only the contribution of an ionization channel. Thus, the cross sections are larger, and the dominating capture process, e.g., positronium production, produces the maximum at much lower energy than the ionization process, e.g., the data of Knudsen et al. [49] and Jacobsen et al. [50]. We can see in Figure 4 that for the positron impact, our three-body CTMC cross sections are in much better agreement with the experimental data than what was found for the electron impact. This is true for both the ionization channel and total electron loss, including both the ionization and positronium formation channels. As for the electron impact, in Figure 4, we also show the scaled experimental ionization cross sections of an atomic hydrogen atom by positron impact [51]. The scaled ionization cross sections are in agreement with both our CTMC results and the experimental data, especially above 50 eV.

In Figure 5, our present three-body CTMC data for the positron impact are compared with a variety of other theoretical models, such as the convergent-close-coupling model [52], the two-center close-coupling model [53], and the results published in Ref. [54], which use two models. One model, denoted by CPE and shown by the blue curves in Figure 5, assumes that when the scattered positron is faster than the ejected electron, the positron moves in the field of the neutral atom, while the electron moves in the field of the positive ion, but when the ejected electron is faster than the scattered positron, the electron moves in the field of the positron and the remaining positive ion, while the positron moves in the field of the positive ion. The model uses simple Coulomb and plane waves for both the initial- and final-state channels. The other model, denoted by CCA and shown by green curves in Figure 5, assumes that in the final state of the system, both the scattered positron and the ejected electron move in the field of the positive ion and are described by Coulomb wavefunctions. The CCA model and the two-center close-coupling model [53] mimic the ionization cross sections, while the CPE and the convergent-close-coupling model [52] show the total electron loss cross sections.

Next, we will show results for single differential cross sections at a 250 eV impact energy. We selected this energy for comparison because, for intermediate energies, our three-body CTMC results are in reasonable agreement with the experimental data for both the electron and positron impact. In Figure 6, the energy distributions of the emitted electron and the scattered projectiles for the 250 eV positron (red curve) and electron (blue curve) impact are compared with available experimental data. Note that the scattered projectile data are shown as a function of the impact energy, E_o_, minus the ionization potential, IP, minus the scattered projectile energy, E_scatterd_. Doing so yields identical results with the ejected electron distributions. Also seen are nearly identical results for positron and electron impact and very good agreement with experimentally measured energy distributions [55,56]. The reader will note that no experimental data are shown for energies larger than half of the impact energy, as in this region, the scattered projectile contribution begins to dominate. Also note that the 200 eV data were scaled to 250 eV using a Bethe scaling technique, e.g., σ (250 eV) = σ (200 eV) [ln T_250_/T_250_]/[ln T_200_/T_200_], where T is the impact energy in atomic units.

In Figure 7, angular distributions for the ejected electron and the scattered projectile are shown. Again, the impact energy is 250 eV, and the positron and electron results are shown in red and blue, respectively. We are not aware of any experimental data that can be used for comparison, but note that for projectile scattering, there is no difference associated with the sign of the projectile charge. However, for the target electron emission, there is a clear difference. In the forward direction, the emission is more probable for the positron impact, whereas in the backward direction, it is more probable for the electron impact.

In Figure 8 and Figure 9, the average impact parameters as a function of the energy and angle are shown. As a function of energy, the average impact parameters for the positron and electron impacts are identical for ejection energies smaller than half the impact energy, i.e., when the scattered projectile is faster than the ejected electron. But, when the scattered projectile is slower than the ejected electron, the average impact parameter for the positron impact becomes increasingly larger than for the electron impact. We also note that electron ejection occurs for collisions at larger distances than is the case for projectile scattering.

On a relative scale, impact parameters for projectile scattering in the extreme forward direction are large, whereas in the backward direction, they are quite small. And there is little or no difference depending on the sign of the projectile charge. However, there are large differences in the average impact parameters for the electron emission. As was seen when comparing differential cross sections, in the forward direction, the values are bigger for the positron impact, whereas in the backward direction, they are larger for the electron impact.

## 3. Method and Theory

In our CTMC model, the three particles are the projectile (P), one atomic active target electron (e), and the remaining target ion (T, the H_2_^+^ ion). Figure 10 shows the relative position vectors of the three-body collision system.

The three particles are characterized by their masses and charges. The Lagrangian for the three particles can be written as
(1)$$L=L_{K}-L_{V}$$
where
(2)$$L_{K}=\frac{1}{2}m_{P}{\dot{\vec{r}}}_{P}^{2}+\frac{1}{2}m_{e}{\dot{\vec{r}}}_{e}^{2}+\frac{1}{2}m_{T}{\dot{\vec{r}}}_{T}^{2}$$
and
(3)$$L_{V}=\frac{Z_{P}Z_{e}}{\left|{\vec{r}}_{P}-{\vec{r}}_{e}\right|}+\frac{Z_{P}Z_{T}}{\left|{\vec{r}}_{P}-{\vec{r}}_{T}\right|}+\frac{Z_{e}Z_{T}}{\left|{\vec{r}}_{e}-{\vec{r}}_{T}\right|}$$

$\vec{r}$, *Z* and *m* are the position vector, the charge, and the mass of the noted particle, respectively. In the following, the lower indexes, *P*, *e*, *T*, are denoted as the projectile, the target electron, and the target nucleus, respectively. Then, the equations of motion can be derived as
(4)$$\frac{d}{dt}\frac{\partial L}{\partial {\dot{q}}_{i}}=\frac{\partial L}{\partial q_{i}}\;\left(i=p,e,T\right)$$

In the present CTMC approach, Newton’s classical nonrelativistic equations of motion for a three-body system were solved numerically for a statistically large number of trajectories. All the forces acting among the particles were taken to be pure Coulombic ones. Therefore, Equation (4) can be written as
(5)$$m_{i}\frac{d^{2}{\vec{r}}_{i}}{dt^{2}}=\sum_{\begin{array}{l} j=1\\ j\neq i \end{array}}^{3}Z_{i}Z_{j}\frac{{\vec{r}}_{i}-{\vec{r}}_{j}}{{\left|{\vec{r}}_{i}-{\vec{r}}_{j}\right|}^{3}}\;\left(i=p,e,T\right)$$

Introducing the relative position vectors $\vec{A}={\vec{r}}_{e}-{\vec{r}}_{T}$, $\vec{B}={\vec{r}}_{T}-{\vec{r}}_{P}$, and $\vec{C}={\vec{r}}_{P}-{\vec{r}}_{e}$ in such a way that $\vec{A}+\vec{B}+\vec{C}=\vec{0}$ (see Figure 10), we can write
(6)$$m_{P}{\ddot{\vec{r}}}_{P}=\frac{Z_{P}Z_{T}}{B^{3}}\;\vec{B}+\frac{Z_{P}Z_{T}}{C^{3}}\vec{C}$$
(7)$$m_{e}{\ddot{\vec{r}}}_{e}=\frac{Z_{e}Z_{T}}{A^{3}}\;\vec{A}-\frac{Z_{P}Z_{e}}{C^{3}}\vec{C}$$
(8)$$m_{T}{\ddot{\vec{r}}}_{T}=\frac{Z_{P}Z_{T}}{B^{3}}\;\vec{B}-\frac{Z_{e}Z_{T}}{A^{3}}\vec{A}$$

After some elementary calculus, Equations (6)–(8) are reduced to the following two ones:(9)$$\ddot{\vec{A}}=\left[\frac{(N_{2}+N_{3})Z_{2}Z_{3}}{{\left|\vec{A}\right|}^{3}}+\frac{N_{2}Z_{1}Z_{2}}{{\left|\vec{A}+\vec{B}\right|}^{3}}\right]\vec{A}+\left[\frac{N_{2}Z_{1}Z_{2}}{{\left|\vec{A}+\vec{B}\right|}^{3}}-\frac{N_{3}Z_{1}Z_{3}}{{\left|\vec{B}\right|}^{3}}\right]\vec{B}$$
(10)$$\ddot{\vec{B}}=\left[-\frac{N_{3}Z_{2}Z_{3}}{{\left|\vec{A}\right|}^{3}}+\frac{N_{1}Z_{1}Z_{2}}{{\left|\vec{A}+\vec{B}\right|}^{3}}\right]\vec{A}+\left[\frac{N_{1}Z_{1}Z_{2}}{{\left|\vec{A}+\vec{B}\right|}^{3}}+\frac{(N_{1}+N_{3})Z_{1}Z_{3}}{{\left|\vec{B}\right|}^{3}}\right]\vec{B}$$

These differential equations are integrated with respect to time as an independent variable by using the standard Runge–Kutta method for a given set of initial conditions. Equations (9) and (10) contain 12 coupled first-order differential equations. Therefore, we need to consider and specify 12 initial values of the initial conditions. These are the coordinates and the velocities of an internal motion of the (T,e) atomic system and the relative projectile electron/positron–atomic center-of-mass motion. Let the origin of our coordinate system in the laboratory frame be the center of mass of the target, and let the z-axis be parallel to the velocity vector of the projectile (see Figure 10). The initial relative motion is specified by the velocity of the projectile and the distance between the projectile and the atomic center of mass:(11)$$\vec{R}=\left(\begin{array}{c} 0\\ b\\ -\sqrt{R^{2}-b^{2}} \end{array}\right)$$
(12)$$\dot{\vec{R}}=\left(\begin{array}{c} 0\\ 0\\ v_{P} \end{array}\right)$$

During our CTMC simulations, $v_{P}$ is fixed. The impact parameter *b* (see Figure 10) is chosen so that it reproduces a uniform flux of incident particles. Apart from elastic collisions, we can determine a maximum value of the impact parameter, *b_max_*, in such a way that by applying impact parameters above *b_max_* in the CTMC calculations, the probabilities of the investigated processes are zero or negligibly small. The initial distance, *R*, between the projectile electron/positron and target atom is chosen at sufficiently large internuclear separations, where the projectile electron/positron and target atom interactions are negligible. In practice, we can select R as *R* = (4,5) *b*_max_ *Z_P_*.

The initial electronic state of the target atom is obtained from the microcanonical distribution.

The total and single differential cross sections can be calculated by
(13)$$\sigma =\frac{2\pi b_{max}}{T_{N}}\sum_{j}b_{j}^{\left(i\right)}\;$$
(14)$$\frac{d\sigma }{dE}=\frac{2\pi b_{max}}{N_{tot}\Delta E}\sum_{i=1}^{N_{t}}b_{i}$$
(15)$$\frac{d\sigma }{d\Omega }=\frac{2\pi b_{max}}{N_{tot}\Delta \Omega }\sum_{i=1}^{N_{t}}b_{i}$$
where *T_N_* is the total number of trajectories calculated for impact parameters less than *b_max_*, $N_{t}\;$ is the number of trajectories that satisfy the criteria for ionization, and $b_{i}$ is the actual impact parameter for the trajectory corresponding to the ionization process under consideration in the energy interval Δ*E* and the emission angle interval ΔΩ of the electron. The statistical error for a given measurement has the form
(16)∆σ=σTN−TN(i)TNTN(i)12

## 4. Conclusions

We have presented studies of total ionization and charge exchange cross sections in collisions between electron and positron impact with molecular hydrogen target. The calculations were performed classically using a three body CTMC approximation where the H_2_ target is described using a central core and a single active electron approximation. We found that this model describes the positron impact total ionization cross sections reasonably well. The same is true for intermediate energy electron impact. But, at low electron impact energies this model overestimates the cross section and at high electron impact energies it underestimates the cross section.

In the intermediate region where good agreement with experimental total ionization cross sections was found, excellent agreement for singly differential energy distributions was also found and no differences associated with the sign of the projectile charge were predicted. No differences for angular scattering were also predicted. However, the angular distributions for electron ejection by positrons and electrons are predicted to be noticeably different.

Impact parameter information for total and differential ionization were also shown. For total ionization the most notable feature is a shift toward smaller impact parameters with increasing impact energy. For differential ionization, the average impact parameters for positron and electron impact were quite similar for the scattered projectile and for lower energy electron emission but quite different for higher energy emission and for all ejection angles. We also have presented, in comparison, the scaled experimental atomic cross sections for both electron and positron impact. We found that the scaled cross sections are in very good agreement with both our CTMC results and with the available experimental data.

## Figures and Tables

**Figure 1 ijms-25-03410-f001:**
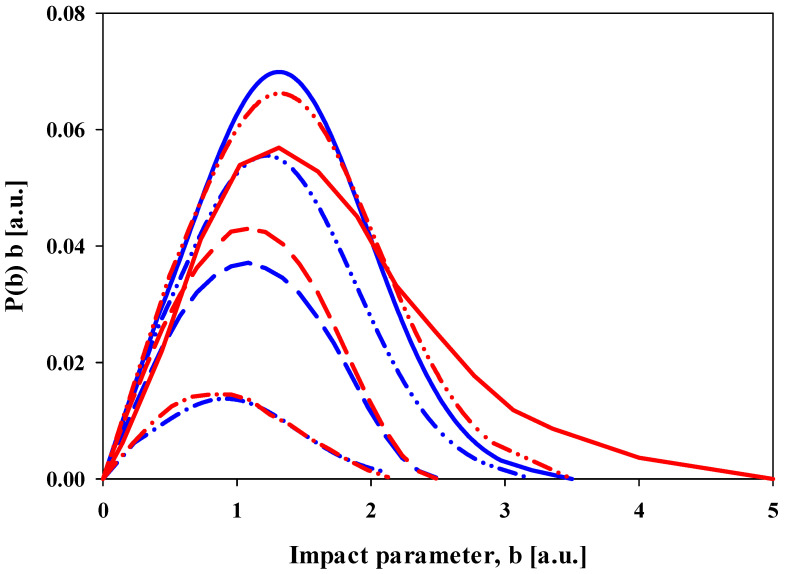
Present CTMC results of the target ionization probabilities in e^−^ + H_2_ (blue lines) and e^+^ + H_2_ (red lines) collisions as a function of impact parameter. Solid line: 50 eV impact energy; double-dotted-dashed line: 100 eV impact energy; dashed line: 250 eV impact energy; dotted-dashed line: 1000 eV impact energy.

**Figure 2 ijms-25-03410-f002:**
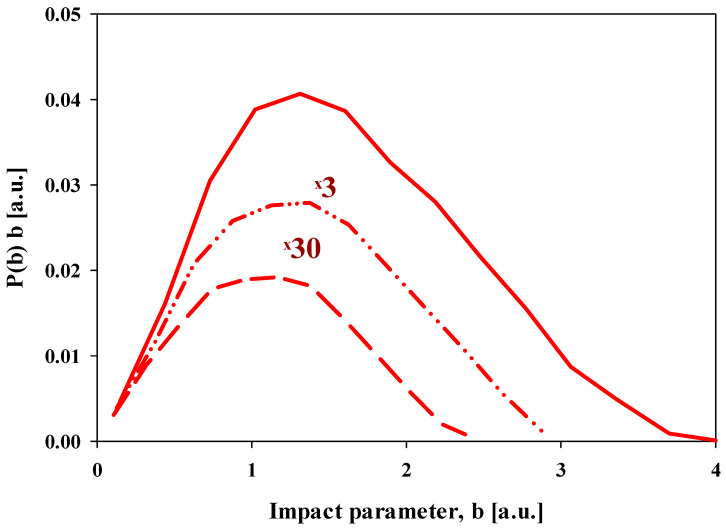
Present CTMC results of the charge exchange probabilities in e^+^ + H_2_ collisions as a function of impact parameter. Solid line: 50 eV impact energy; double-dotted-dashed line: 100 eV impact energy, the original data were multiplied by 3; dashed line: 250 eV impact energy, the original data were multiplied by 30.

**Figure 3 ijms-25-03410-f003:**
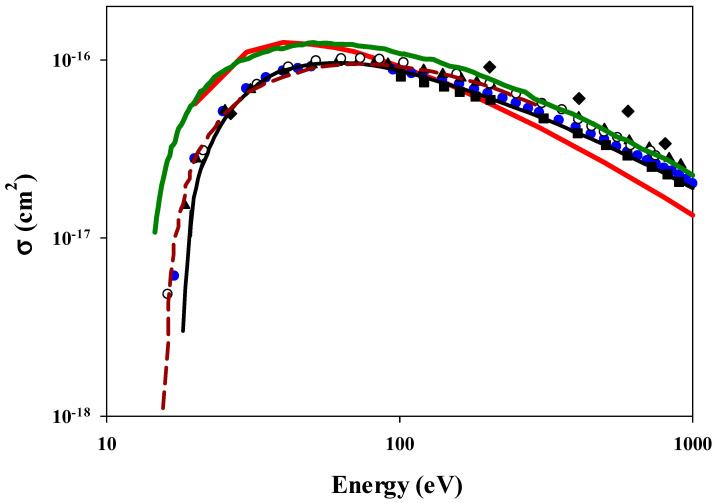
Total ionization cross section as a function of the impact energy in collision between an electron and a hydrogen molecule. Solid-red line: present CTMC results. Theories: solid-black line: Khare and Padalia [38], dark-red-dashed line: convergent close-coupling method within the fixed-nuclei approximation [39,40]. Experimental cross sections: open circle: Tate and Smith [41]; solid diamond: Harrison [41]; solid triangle: Rapp and Englander-Golden [43], solid square: Schram et al. [44]; solid circle: Straub et al. [45]; dark-green line: experimental ionization cross sections [46] between an electron and atomic hydrogen multiplied by 2.

**Figure 4 ijms-25-03410-f004:**
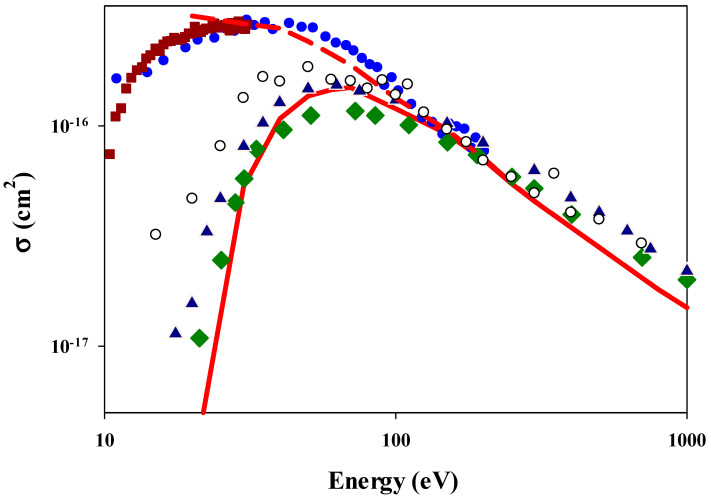
Total ionization cross sections as a function of the impact energy in collision between positron and hydrogen molecule. Solid-red line: present CTMC results for ionization channel, dashed-red line: present CTMC results for the total electron loss of the target, i.e., the sum of the ionization and charge exchange channels. Experimental total electron loss cross sections of the target: solid circle: Fromme et al. [47]; solid square: Moxom et al. [48]. Experimental ionization cross cross sections of the target: solid triangle: Knudsen, et al. [49], solid diamond: Jacobsen et al. [50]. Note that the Fromme et al. and Moxom et al. data include and, at lower energies, are dominated by positronium production, i.e., the charge exchange channel, open circles: experimental ionization cross sections [51] between a positron and atomic hydrogen multiplied by 2.

**Figure 5 ijms-25-03410-f005:**
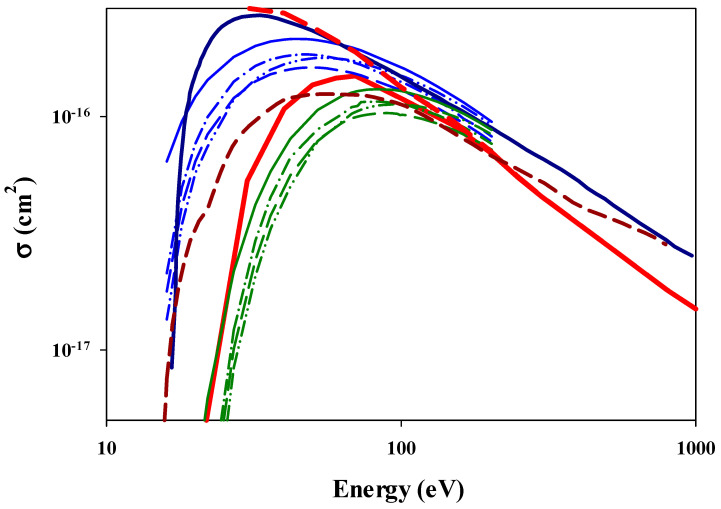
Total ionization cross sections as a function of the impact energy in collision between positron and hydrogen molecule. Solid-red line: present CTMC results for ionization channel, dashed red line: present CTMC results for the total electron loss of the target, i.e., the sum of the ionization and charge exchange channels. Dark blue solid line: convergent-close-coupling model [52], dark-red-dashed line: two-center close-coupling method [53], blue-double-dotted-dashed line: CCP-WANG model in Ref. [54]; solid-blue line: CCP-CMV model in Ref. [54]; blue-dotted-dashed line: CCP-HUZ3 model in Ref. [54]; dashed-blue line: CCP-HUZ1 model in Ref. [54]; green-double-dotted-dashed line: CCA-WANG model in Ref. [54]; solid-green line: CCA-CMV model in Ref. [54]; green-dotted-dashed line: CCA-HUZ3 model in Ref. [54]; dashed-green line: CCA-HUZ1 model in Ref. [54].

**Figure 6 ijms-25-03410-f006:**
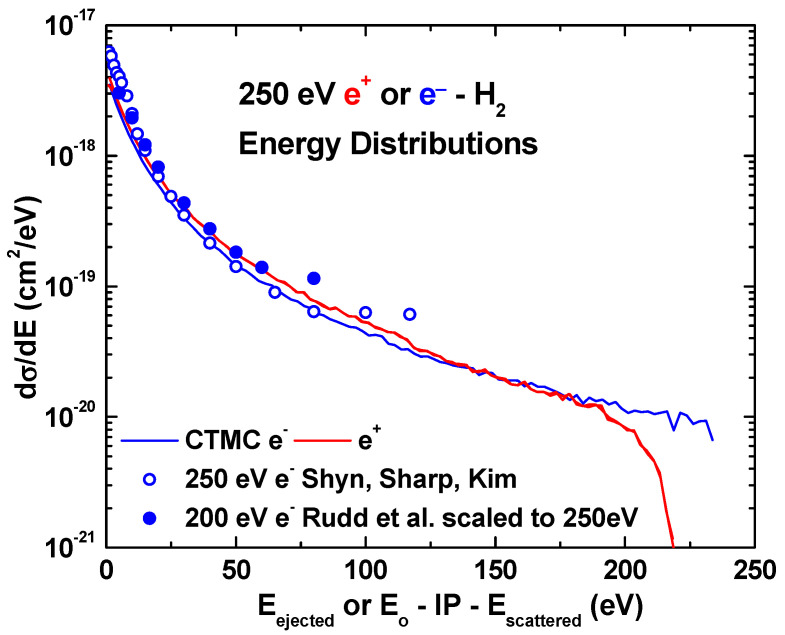
Energy distribution of the emitted electron and scattered projectiles at 250 eV projectile impact. The scattered projectile energy scale is the difference between the scattered projectile energy, E_scattered_, and the available energy, E_o_–IP, where E_o_ is the impact energy and IP is the ionization energy. See text for details. Present three-body CTMC results, positron (red curve), electron (blue curve); experimental data of Shyn, Sharp, and Kim [55] (blue open circles) and Rudd et al. [56] (blue filled circles).

**Figure 7 ijms-25-03410-f007:**
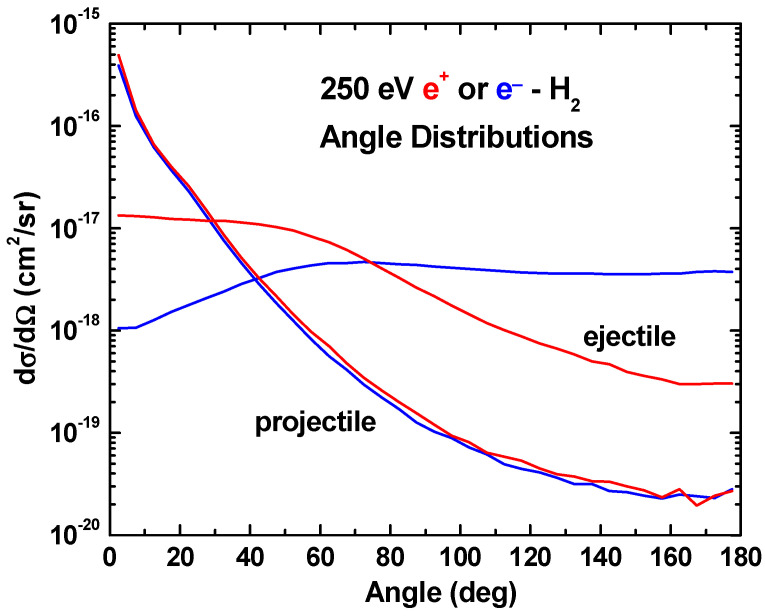
Our three-body CTMC predictions for the angular distributions of the emitted electron and scattered projectiles for 250 eV positron (red curves) and electron (blue curves) impact.

**Figure 8 ijms-25-03410-f008:**
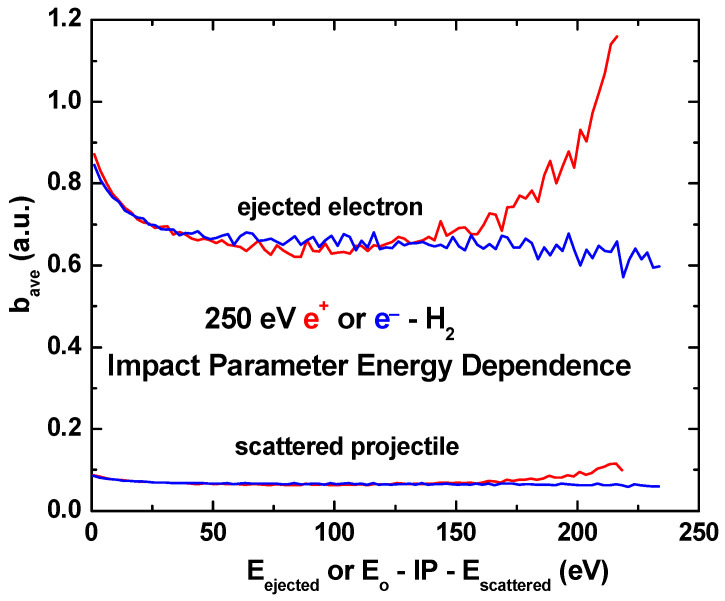
Average impact parameters as a function of energy for 250 eV positron (red curves) and electron (blue curves) impact. Energy scale same as in Figure 5.

**Figure 9 ijms-25-03410-f009:**
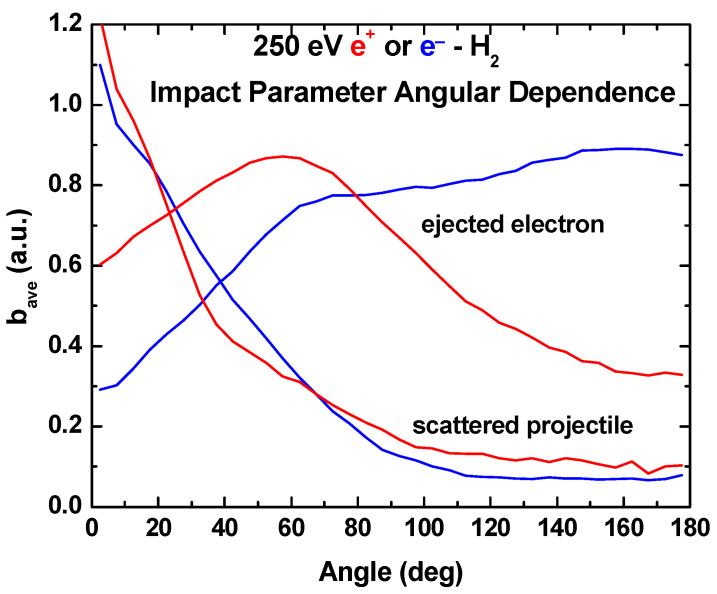
Average impact parameters as a function of angle for 250 eV positron (red curves) and electron (blue curves) impact.

**Figure 10 ijms-25-03410-f010:**
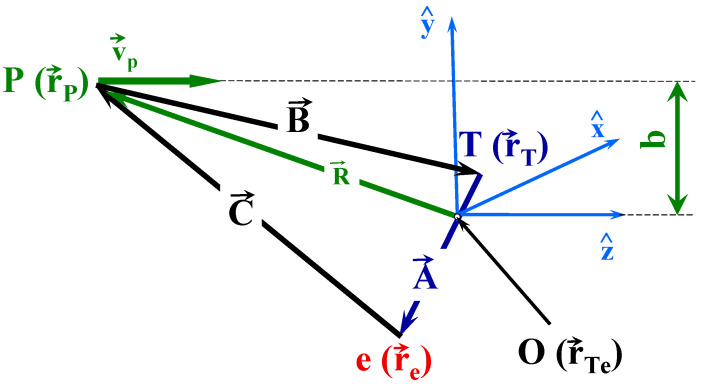
The relative position vectors of the particles involved in three-body collisions. $\vec{A}={\vec{r}}_{e}-{\vec{r}}_{T}$, $\vec{B}={\vec{r}}_{T}-{\vec{r}}_{p}$ and $\vec{C}={\vec{r}}_{p}-{\vec{r}}_{e}$, in such way that $\vec{A}+\vec{B}+\vec{C}=0$. Also, ${\vec{r}}_{\mathrm{Te}}$ is the position vector of the center-of-mass of the target system, and *b* is the impact parameter.

## Data Availability

The data that support the findings of this study are available from the authors upon reasonable request.

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
