# Peer review of "Ionization Cross Sections of Hydrogen Molecule by Electron and Positron Impact"

_ijms, 2024, doi:10.3390/ijms25063410_

Round 1
Reviewer 1 Report
Comments and Suggestions for Authors
See uploaded file.

Author Response
Reviewer 1
We would like to thank the referees for the critical reading of our manuscript and for the positive comments.
Q1: The understanding of the ionization and charge exchange processes in ion-atom and ion-molecule collisions is of fundamental interest …” In the paper, the authors deal with electron – molecular collisions. Why does the introduction talk about ion-atomic and ion-molecular collisions.
A1: The introduction was modified according to the referee request.
Q2: a) Page 4. “These are the coordinates and the velocities of an internal motion of (T,e) atomic system and the relative projectile ion –atomic center-of-mass motion.” b) Page 5. “The initial distance, R, between the projectile ion and target atom is chosen at sufficiently large internuclear separations, where the projectile ion and target atom interactions are negligible.” Probably there should be “projectile electron/positron” instead of “projectile ion”
A2: We follow the referee suggestions and modified the manuscript accordingly.
Q3: The effective binding energy used is 0.567 a.u. » 15.4 eV, i.e. is equal to ionization energy of hydrogen molecules, and it looks rather reasonable. At the same time the effective core charge used is 1.165, i.e. >1. Why didn't the authors use the value Zeff = 1? It would seem that this is a more natural value for the model under consideration.
A3: In principle the active electron feels for both proton charge in the real hydrogen molecule. Therefore, the effective core charge was selected higher than 1 because mimicking the influence of the other proton positive charge.
Q4: Caption to figure 4. “… solid black line: theory …” There is no “solid black line” in the figure. There is dashed black line.
A4: Thanks to the referee to call our attention for this graphical error. It is corrected.
Q5: Figure 5. Results of calculations for the total electron loss (dashed red line) are presented for energies > 20 eV, while experimental data starts at 10 eV. Why didn't the authors provide the results of calculations in the energy range 10 eV- 20 eV
A5: As it is known, the validity of the standard CTMC calculations is less and less at lower impact energies. Therefore, we did not calculate the cross sections below 20 eV.
Q6a: Caption to figure 5 and text on page 9. In the caption: “…dashed red line: present CTMC results for the total electron loss of the target, i.e. the sum of the ionization and charge exchange channels. Experimental total electron loss cross sections of the target: solid circle: Fromme et al. [39]; solid square: Moxom et al. [40].” It follows from this part of the caption that the term “total electron loss cross section” designate the sum of “ionization and charge transfer channels”. And in papers by Fromme et al. [39] and Moxom et al. [40] exactly “total electron loss cross section” was measured. Further, in the caption there is a phrase “Note that the Fromme et al. and Moxom et al. data include and, at lower energies, are dominated by positronium production …” The terms “positronium production” and “positronium formation” are used also several times in the text on page 9. For example, “The differences are because the cross sections of Fromme et al. [39], and Moxom et al. [40] contain both ionization and positronium production processes …”
It becomes unclear whether the terms “charge exchange channel” and “positronium production channel” mean the same process or whether they are different processes.
A6a: The “charge exchange channel” and “positronium production channel” is the same. We corrected the text.
Q6b: For the convenience of readers (to avoid confusion), authors should write down all the processes under consideration in the form of formulas at the beginning of the article (for example, H2 + e = H2 + + e + e)
A6b: We think that the definition of ionization is straightforward, and it is not necessary to include the “definition” of ionization channel in this manuscript.
Q7: Calculated cross sections are compared with the scaled cross sections obtained experimentally for an atomic hydrogen target. What are the physical grounds for this comparison? The authors make a formal comparison without explaining the reasons why this comparison is necessary.
A7: It is not necessary, sure, we agree with the referee. However, as the hydrogen atom can maybe treat as two individual hydrogen atom, we think it is an addition to the results we obtained.
Reviewer 2 Report
Comments and Suggestions for Authors
Although a long history and there are different approaches to the problem H_2-e,e^+ collisions and it's still impossible to point out today the versatile approach to the problem. The authors took a three body classical trajectory Monte Carlo approximation to deal with the problem for impact energies between 20 and 1000eV. The H_2 molecule is modeled by a hydrogen-type atom with one active electron bound to a central core of effective charge with an effective binding energy. The results for the total cross section for positron impact agree well with experimental data but for electron impact threre are disagreement in the regions of energies more then 400eV and less then 80eV. Thus, the results obtained is one more attempt to apply a classical approach to electron collisions.
In the present work an ensemble of 10^6 trajectories was used but it is unclear how the results were sensitive to the amount of trajectories and what is more important how to include in a classical approach the electron exchange in the region of low energies.
Author Response
Reviewer 2
We would like to thank the referees for the critical reading of our manuscript and for the positive comments.
Q1: In the present work an ensemble of 10^6 trajectories was used but it is unclear how the results were sensitive to the amount of trajectories and what is more important how to include in a classical approach the electron exchange in the region of low energies.
A1: The accuracy of the Monte Carlo simulations strongly depends on the used number of primary trajectories. Please see the Eq. 16 in the manuscript. The more primary trajectories give more accurate results.
The definition of the final channels including the charge exchange channel id based on the checking the two body relative energies. It is independent from the primary energy. However, we note that the validity of the standard CTMC calculations is less and less at lower impact energies for charge exchange channel.
Reviewer 3 Report
Comments and Suggestions for Authors
The manuscript describes computational modelling of electron and positron collisions with molecular hydrogen. The authors used a three-body classical trajectory Monte Carlo approximation is applied to model the collision system. The H2 molecule was described by a very crude model – as a quasi-one-electron atom. For electron impact, ionization cross sections have been calculated. For positron impact, ionization and positronium formation cross sections have been calculated. Comparisons with the experiment and other calculations have been presented. The authors found a reasonable agreement with the experiment and other accurate calculations. This is an interesting outcome, given a very simplified model for the H2 molecule and the classical nature of the calculations.
In general, the manuscript is suitable for publication in this journal.
There are some issues that I recommend to deal with before the formal acceptance.
The main issue that I see is related to the use of a very simplified model for H2 molecules – a quasi-one-electron model. Such a simple model allows for a fully quantum treatment for both electron and positron scattering – as it was done for hydrogen atoms and alkali atoms (Li, Na,…). Did the authors apply their technique to the hydrogen atom? It is where very accurate calculations exist for both electron and positron scattering. This should be the natural testing ground for the CTMC method.
Another point is that if the authors aim to model other molecules as simplified quasi-one-electron systems then why bother with the CTMC method as the fully quantum techniques can be applied?
The authors need to be more clear on the motivation for their work. Ideally, it would be better to use a more accurate model for H2 molecules.
Below are some more specific comments.
1. page 1, Introduction, 5th sentence
Authors wrote: “A theoretical method that accounts for all these forces is the Classical Trajectory Monte Carlo (CTMC) method.”
Surely, there are forces and interactions that classical physics does not account for: the electron exchange is an example.
2. page 3, Eq. (1)
The authors wrote: “The Lagrange equation for the three particles can be written as: ”
Eq. (1) is not the Lagrange equation.
3. page 3, Eq. 4
The authors wrote: “Than the equations of motion can be calculated as:”
It is better to say that the equations are derived.
Change Than to Then
4. page 8, Fig. 4
The most accurate ab initio cross sections from the CCC method [PRL 116(2016)233201, PRA95(2017)022708] are not presented.
Instead, a comparison is made with semi-empirical calculations.
The figure, in its present form, is misleading: it does not represent the current state of the field.
5. page 10, Fig. 6
The CCC calculations were done in the two-centre approach: both electron loss and direct ionization cross sections are available, see Phys. Rev. A 92, 032707.
Comments on the Quality of English Language
mostly OK, but some minor editing is required.
Author Response
Reviewer 3
We would like to thank the referees for the critical reading of our manuscript and for the positive comments. We appreciate it very much for the suggestions to improve the manuscript. We accept and follow all suggestions.
GQ1a: The main issue that I see is related to the use of a very simplified model for H2 molecules – a quasi-one-electron model. Such a simple model allows for a fully quantum treatment for both electron and positron scattering – as it was done for hydrogen atoms and alkali atoms (Li, Na,…). Did the authors apply their technique to the hydrogen atom? It is where very accurate calculations exist for both electron and positron scattering. This should be the natural testing ground for the CTMC method.
GA1a: Yes, we have a large number of unpublished data both for electron and positron impact calculations on hydrogen atom at 50 eV and 100 eV. We just plotted together our CTMC results with the CCC results of the Australian group. We have very good agreement for positron impact, and good agreement for electron impact. The electron impact data is a bit higher at lover impact energy while at 100 eV the agreement between CCC and CTMC is very excellent.
GQ1b: Another point is that if the authors aim to model other molecules as simplified quasi-one-electron systems then why bother with the CTMC method as the fully quantum techniques can be applied?
GA1b: The referee is right, the system is reduced to 3 body problem, however the initial conditions are obtained, based on the experimental and other calculations. The simplicity is the advantage. The calculations are straightforward.
GQ1c: The authors need to be more clear on the motivation for their work. Ideally, it would be better to use a more accurate model for H2 molecules.
GA1c: Partly please see the answer A1b. We would have liked to test, that this simple model how accurately can describe the more complex system. It seems to be we can have a very close resuls with the experimental data. It can give quick and fast results.
Q1: page 1, Introduction, 5th sentence
Authors wrote: “A theoretical method that accounts for all these forces is the Classical Trajectory Monte Carlo (CTMC) method.”
Surely, there are forces and interactions that classical physics does not account for: the electron exchange is an example.
A1: The electron exchange process also can be treated in a classical manner, however in this approximation it does not play a role.
Q2: page 3, Eq. (1)
The authors wrote: “The Lagrange equation for the three particles can be written as: ”
Eq. (1) is not the Lagrange equation.
A2: It is corrected.
Q3: page 3, Eq. 4
The authors wrote: “Than the equations of motion can be calculated as:”
It is better to say that the equations are derived.
Change Than to Then
A3: It is corrected.
Q4: page 8, Fig. 4
The most accurate ab initio cross sections from the CCC method [PRL 116(2016)233201, PRA95(2017)022708] are not presented.
Instead, a comparison is made with semi-empirical calculations.
The figure, in its present form, is misleading: it does not represent the current state of the field.
A4: Thanks to the referee to send these references. We show now these results also. Just we note that for positron impact we referred in the original manuscript the CCC results as ref 44.
Q5: page 10, Fig. 6
The CCC calculations were done in the two-centre approach: both electron loss and direct ionization cross sections are available, see Phys. Rev. A 92, 032707.
A5: Again thanks to the referee to send us this reference. We cite it in the manuscript.